# Multi-armed Bandit Requiring Monotone Arm Sequences

**Ningyuan Chen**
Rotman School of Management, University of Toronto
105 St George St, Toronto, ON, Canada
`ningyuan.chen@utoronto.ca`

## Abstract

In many online learning or multi-armed bandit problems, the taken actions or pulled arms are ordinal and required to be monotone over time. Examples include dynamic pricing, in which the firms use markup pricing policies to please early adopters and deter strategic waiting, and clinical trials, in which the dose allocation usually follows the dose escalation principle to prevent dose limiting toxicities. We consider the continuum-armed bandit problem when the arm sequence is required to be monotone. We show that when the unknown objective function is Lipschitz continuous, the regret is $O(T)$. When in addition the objective function is unimodal or quasiconcave, the regret is $\tilde{O}(T^{3/4})$ under the proposed algorithm, which is also shown to be the optimal rate. This deviates from the optimal rate $\tilde{O}(T^{2/3})$ in the continuous-armed bandit literature and demonstrates the cost to the learning efficiency brought by the monotonicity requirement.

## 1 Introduction

In online learning problems such as the multi-armed bandits (MAB), the decision maker chooses from a set of actions/arms with unknown reward. The goal is to learn the expected rewards of the arms and choose the optimal one as much as possible within a finite horizon. The framework and its many variants have been successfully applied to a wide range of practical problems, including news recommendation [31], dynamic pricing [9] and medical decision-making [8].

Popular algorithms such as UCB [4, 5] and Thompson sampling [3, 34] typically explore the arms sufficiently and as more evidence is gathered, converge to the optimal arm. These algorithms are designed to handle the classic MAB problems, in which the arms are discrete and do not have any order. However, in some practical problems, the arms are ordinal and even continuous. This is the focus of the study on the so-called continuum-armed bandit [2, 26, 6, 11, 38]. By discretizing the arm space properly, algorithms such as UCB and Thompson sampling can still be applied. Therefore, the continuous decision space has readily been incorporated into the framework.

Another practical issue, which is not studied thoroughly but emerges naturally from some applications, is the requirement that the action/arm sequence is has to be monotone. Such a requirement is usually imposed due to external reasons that cannot be internalized to be part of the regret. We give two concrete examples below.

**Dynamic pricing**. When launching a new product, how does the market demand reacts to the price is typically unknown. Online learning is a popular tool for the firm to learn the demand and make profits simultaneously. See [21] for a comprehensive review. In this case, the arms are the prices charged by the firm. However, changing prices arbitrarily over time may have severe adverse effects. First, early adopters of the product are usually loyal customers. Price drops at a later stage may cause a backlash in this group of customers, whom the firm is the least willing to offend. Second, customers

35th Conference on Neural Information Processing Systems (NeurIPS 2021).

are usually strategic, as studied extensively in the literature [39, 32, 44, 31, 16]. The possibility of a future discount may make consumers intentionally wait and hurt the revenue. A remedy for both issues is a *markup* pricing policy, in which the firm increases the price over time. This is particularly common for digital services growing rapidly: both Dropbox and Google Drive went through the phases from free, to low fees, to high fees. In a slightly different context, markup pricing is often used in successful crowdfunding campaigns: to reward early backers, the initial products may be given for free; later after some goals are achieved, backers may get the product at a discount; after the release, the product is sold at the full price. It is not a good idea to antagonize early backers by lowering the price later on. In the language of multi-armed bandit, it is translated to the requirement that the pulled arms in each period are increasing over time.

**Clinical trials**. Early stage clinical trials for the optimal dose are usually conducted in Phase I/II trials sequentially. For Phase I, the toxicity of the doses is examined to determine a safe dose range. In this phase, the dose allocation commonly follows the "dose escalation" principle [30] and the "3+3 design" is the most common dose allocation approach. For Phase II, the efficacy of the doses are tested within the safe dose range (by random assignment, for example). Recently, especially in oncology, there are proposals to integrate both phases [42] which is called "seamless Phase I/II trials." In this case, dose escalation has to be observed when looking for the most efficacious dose. Our study is thus motivated by seamless trials. In the online learning framework, the dosage is treated as arms and the efficacy is the unknown reward. Multi-armed bandit has been applied to the dose allocation to identify the optimal dose [41, 7]. In particular, after a dose has been assigned to a cohort of patients, their responses are recorded before the next cohort is assigned a higher dose. If any of the patients experiences dose limiting toxicities (DLTs), then the trial stops immediately. The dose escalation principle requires the allocated doses to be increasing over time. A recent paper incorporates the principle in the design of adaptive trials [13].

To accommodate the above applications, in this paper, we incorporate the requirement of monotone arm sequences in online learning, or more specifically, the continuum-armed bandit framework. We consider an unknown continuous function $f(x)$ and the decision maker may choose an action/arm $X_t$ in period $t$. The observed reward $Z_t$ is a random variable with mean $f(X_t)$. The requirement of monotone arm sequences is translated to $X_1 \leq X_2 \leq \cdots \leq X_T$, where $T$ is the length of the horizon. The goal of the decision maker is to minimize the regret $T \max f(x) - \sum_{t=1}^{T} \mathrm{E}[f(X_t)]$.

**Main results**. Next we summarize the main results of the paper informally. We first show that for a general continuous function $f(x)$, the extra requirement makes learning impossible.

**Result.** *For the class of continuous functions $f(x)$, the regret is at least $T/4$.*

This diverges from the results in the continuum-armed bandit literature. In particular, [26] shows that the optimal regret for continuous functions scales with $T^{2/3}$ without the monotonicity requirement. Therefore, to prove meaningful results, we need to restrict the class of functions $f(x)$. From both the practical and theoretical points of view, we find that assuming $f(x)$ to be unimodal or quasiconcave is reasonable. In practice, for the two applications mentioned above, the revenue function in dynamic pricing is typically assumed to be quasiconcave in the price [47]; for clinical trials, the majority of the literature assumes the dose-response curve to be increasing [10], increasing with a plateau effect [33], or unimodal [45], all of which satisfy the assumption. In theory, with the additional assumption, we can show that

**Result.** *There is an algorithm that has monotone arm sequences and achieves regret $O(\log(T)T^{3/4})$ for continuous and quasiconcave functions $f(x)$.*

Note that the rate $T^{3/4}$ is still different from that in the continuum-armed bandit literature. We then proceed to close the case by arguing that this is the fundamental limit of learning, not due to the design of the algorithm.

**Result.** *No algorithm can achieve less regret than $T^{3/4}/32$ for the instances we construct.*

The constructed instances that are hard to learn (see Figure 2) are significantly different from the literature. In fact, the classic construction of "needle in a haystack" [27] does not leverage the requirement of monotone arm sequences and does not give the optimal lower bound. We prove the lower bound by using a novel random variable that is only a stopping time under monotone arm sequences. By studying the Kullback-Leibler divergence of the probabilities measures involving

Table 1: The *optimal* regret under different settings in the continuum-armed bandit. The regret in [18] is achieved under additional assumptions.

| Paper | Concavity | Quasiconcavity | Monotone Arm Sequences | Regret |
|---|---|---|---|---|
| [26] | N | N | N | $\tilde{O}(T^{2/3})$ |
| This paper | N | N | Y | $O(T)$ |
| [20] | Y | Y | N | $\tilde{O}(T^{1/2})$ |
| [35] | Y | Y | Y | $\tilde{O}(T^{1/2})$ |
| [18] | N | Y | N | $\tilde{O}(T^{1/2})$ |
| This paper | N | Y | Y | $\tilde{O}(T^{3/4})$ |

the stopping time, we are able to obtain the lower bound. We also show numerical studies that complement the theoretical findings.

**Related literature**. The framework of this paper is similar to the continuum-armed bandit problems, which are widely studied in the literature. The earlier studies focus on univariate Lipschitz continuous functions [2, 26] and identify the optimal rate of regret $O(T^{2/3})$. Later papers study various implications including smoothness [6, 15, 24], the presence of constraints and convexity [9, 43], contextual information and high dimensions [38, 12], and so on. The typical reduction used in this stream of literature is to adopt a proper discretization scheme and translate the problem into discrete arms. In the regret analysis, the Lipschitz continuity or smoothness can be used to control the discretization error. Our setting is similar to the study in [26] with quasiconcave objective functions and the additional requirement of monotone arm sequences. The optimal rate of regret, as a result of the difference, deteriorates from $T^{2/3}$ to $T^{3/4}$. This deterioration can be viewed as the cost of the monotonicity requirement.

In a recent paper, [35] study a similar problem while assuming the function $f(x)$ is strictly concave. The motivation is from the notion of "fairness" in sequential decision making [46]. The concavity allows them to develop an algorithm based on the gradient information. The resulting regret is $O(T^{1/2})$, the same as continuum-armed bandit with convexity/concavity [20, 1]. Comparing to this set of results, the strong concavity prevents the deterioration of regret rates under the monotonicity requirement. In contrast, [18, 19] show that the same rate of regret $O(T^{1/2})$ can be achieved if the function is not strictly concave, but just unimodal or quasiconcave, under additional technical assumptions. It is worth noting that many results in this paper are discovered independently in a recent paper [25]. The comparison of the regret rates are shown in Table 1.

Other requirements on the arm sequence motivated by practical problems are also considered in the literature. [17, 36, 37] are motivated by a similar concern in dynamic pricing: customers are hostile to frequent price changes. They propose algorithms that have a limited number of switches and study the impact on the regret. Motivated by Phase I clinical trials, [22] study the best arm identification problem when the reward of the arms are monotone and the goal is to identify the arm closest to a threshold. [14] consider the setting when the reward is nonstationary and periodic, which is typical in the retail industry as demand exhibits seasonality.

**Notations.** We use $\mathbb{Z}_+$ and $\mathbb{R}$ to denote all positive integers and real numbers, respectively. Let $T \in \mathbb{Z}_+$ be the length of the horizon. Let $[m] := \{1, \ldots, m\}$ for any $m \in \mathbb{Z}_+$. Let $x^* = \arg\max_{x \in [0,1]} f(x)$ be the set of maximizers of a function $f(x)$ in $[0, 1]$. The indicator function is represented by $\mathbb{1}$.

## 2 Problem Setup

The objective function $f(x) : [0, 1] \to [0, 1]$ is unknown to the decision maker. In each period $t \in [T]$, the decision maker chooses $X_t \in [0, 1]$ and collects a random reward $Z_t$ which is independent of everything else conditional on $X_t$ and satisfies $\mathrm{E}[Z_t|X_t] = f(X_t)$. After $T$ periods, the total (random) reward collected by the decision maker is thus $\sum_{t=1}^{T} Z_t$. Since the decision maker does not know $f(x)$ initially, the arm chosen in period $t$ only depends on the previously chosen arms and observed rewards. That is, for a policy $\pi$ of the decision maker, we have $X_t = \pi_t(X_1, Z_1, \ldots, X_{t-1}, Z_{t-1}, U_t)$, where

$U_t$ is an independent random variable associated with the internal randomizer of $\pi_t$ (e.g., Thompson sampling).

Before proceeding, we first introduce a few standard assumptions.

**Assumption 1.** *The function $f(x)$ is c-Lipschitz continuous, i.e.,*

$$|f(x_1) - f(x_2)| \leq c|x_1 - x_2|,$$

*for $x_1, x_2 \in [0, 1]$.*

This is a standard assumption in the continuum-armed bandit literature [26, 6, 11, 38]. Indeed, without continuity, the decision maker has no hope to learn the objective function well. In this study, we do not consider other smoothness conditions.

**Assumption 2.** *The reward $Z_t$ is $\sigma$-subgaussian for all $t \in [T]$. That is, conditional on $X_t$,*

$$\mathrm{E}[\exp(\lambda(Z_t - f(X_t)))|X_t] \leq \exp(\lambda^2\sigma^2/2)$$

*for all $\lambda \in \mathbb{R}$.*

Subgaussian random variables are the focus of most papers in the literature, due to their benign theoretical properties. See [29] for more details.

We can now define an oracle that has the knowledge of $f(x)$. The oracle would choose arm $x \in \arg\max_{x \in [0,1]} f(x)$ and collect the total reward with mean $T \max_{x \in [0,1]} f(x)$. Because of Assumption 1, the maximum is well defined. We define the regret of a policy $\pi$ as

$$R_{f,\pi}(T) = T \max_{x \in [0,1]} f(x) - \sum_{t=1}^{T} \mathrm{E}[f(X_t)].$$

Note that the regret depends on the unknown function $f(x)$, which is not very meaningful. As a standard setup in the continuum-armed bandit problem, we consider the worst-case regret for all $f$ satisfying Assumption 2.

$$R_\pi(T) := \max_f R_{f,\pi}(T).$$

Next we present the key requirement in this study: the sequence of arms must be *increasing*.

**Requirement 1** (Increasing Arm Sequence). *The sequence $\{X_t\}_{t=1}^T$ must satisfy*

$$X_1 \leq X_2 \leq \cdots \leq X_T$$

*almost surely under $\pi$.*

Note that "increasing" can be relaxed to "monotone" without much effort. For the simplicity of exposition, we only consider the increasing arm sequence.

As explained in the introduction, this is a practical requirement in several applications. In phase I/II clinical trials, the quantity $X_t$ is the dose applied to the $t$-th patient and $f(x)$ is the average efficacy of dose $x$. The dose escalation principle protects the patients from DLTs. More precisely, the doses applied to the patients must increase gradually over time such that the trial can be stopped before serious side effects occur to a batch of patients under high doses.

In dynamic pricing, $X_t$ is the price charged for the $t$-th customer and $f(x)$ is the expected revenue under price $x$. The increasing arm sequence corresponds to a *markup* pricing strategy that is typical for new products. Because of the increasing price, early adopters, who are usually loyal to the product, wouldn't feel betrayed by the firm. Moreover, it deters strategic consumer behavior that may cause a large number of customers to wait for promotions, hurting the firm's profits.

We point out that under Assumption 1 and 2, which typically suffice to guarantee sublinear regret, the regret $R_\pi(T)$ is at least $T/4$ under Requirement 1 for all policies $\pi$.

**Proposition 1.** *For all $\pi$ satisfying Requirement 1, we have $R_\pi(T) \geq T/4$ under Assumptions 1 with $c = 2$ and Assumption 2.*

The result can be easily illustrated by the two functions ($f_1(x)$ in the left and $f_2(x)$ in the right) in Figure 1. Because of Requirement 1 and the fact that $f_1(x) = f_2(x)$ for $x \in [0, 0.5]$, we must have

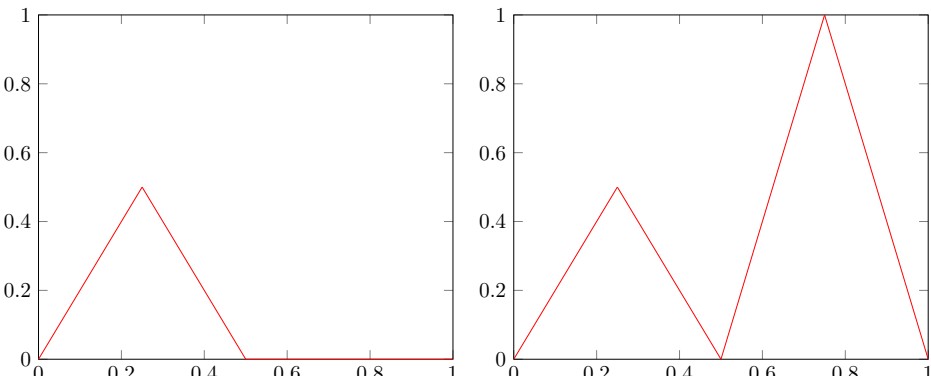

Figure 1: Two functions ($f_1(x)$ in the left and $f_2(x)$ in the right) that any policy cannot differentiate before any $t$ such that $X_t \leq 0.5$ under Requirement 1.

$\sum_{t=1}^{T} \mathbb{1}_{X_t \leq 0.5}$ being equal under $f_1(x)$ and $f_2(x)$ for any given policy $\pi$. Moreover, it is easy to see that

$$R_{f_1,\pi}(T) \geq \frac{T}{2} - \frac{1}{2}\sum_{t=1}^{T} \mathbb{1}_{X_t \leq 0.5}, \quad R_{f_2,\pi}(T) \geq T - \sum_{t=1}^{T} \mathbb{1}_{X_t > 0.5} - \frac{1}{2}\sum_{t=1}^{T} \mathbb{1}_{X_t \leq 0.5}.$$

Recall the fact $\sum_{t=1}^{T} \mathbb{1}_{X_t \leq 0.5} + \sum_{t=1}^{T} \mathbb{1}_{X_t > 0.5} = T$. The above inequalities lead to $R_{f_1,\pi}(T) + R_{f_2,\pi}(T) \geq T/2$, which implies Proposition 1.

Therefore, to obtain sublinear regret, we need additional assumptions for $f(x)$. In particular, we assume

**Assumption 3.** *The function $f(x)$ is quasiconcave, i.e., for any $x_1, x_2, \lambda \in [0, 1]$, we have*

$$f(\lambda x_1 + (1 - \lambda)x_2) \geq \min\{f(x_1), f(x_2)\}.$$

In other words, a quasiconcave function is weakly unimodal and does not have multiple separated local maximums. Therefore, $f_2(x)$ in Figure 1 is not quasiconcave while $f_1(x)$ is. Note that a quasiconcave function may well have a plateau around the local maximum.

We impose Assumption 3 on the function class due to two reasons. First, quasiconcavity is probably the weakest technical assumption one can come up with that is compatible with Requirement 1. It is much weaker than concavity [35], and it is easy to see from Figure 1 that multiple local maximums tend to cause linear regret under Requirement 1. Moreover, unimodal functions are the focus of other studies in the online learning literature [18, 19]. Second, objective functions in practice are commonly assumed to be quasiconcave. For the dynamic pricing example, the revenue function is typically assumed to be quasiconcave in the price [47], supported by theoretical and empirical evidence. In clinical trials, most literature assumes the dose-response curve to be quasiconcave [10, 45, 33].

## 3 The Algorithm and the Regret Analysis

In this section, we present the algorithm and analyze its regret. We first introduce the high-level idea of the algorithm. The algorithm discretizes $[0, 1]$ into $K$ evenly-spaced intervals with end points $\{0, 1/K, \ldots, (K-1)/K, 1\}$, where $K$ is a tunable hyper-parameter. The algorithm pulls arm $k/K$ for $m$ times sequentially for $k = 0, 1, \ldots$, for a hyper-parameter $m$. It stops escalating $k$ after a batch of $m$ pulls if the reward of arm $k/K$ is not as good as that of a previous arm. More precisely, the stopping criterion is that the upper confidence bound for the average reward of arm $k/K$ is less than the lower confidence bound for that of some arm $i/K$ for $i < k$. When the stopping criterion is triggered, the algorithm always pulls arm $k/K$ afterwards. The detailed steps of the algorithm are shown in Algorithm 1.

The intuition behind Algorithm 1 is easy to see. Because of Assumption 3, the function $f(x)$ is unimodal. Therefore, when escalating the arms, the algorithm keeps comparing the upper confidence

---

**Algorithm 1** Increasing arm sequence

---

Input: $K, m, \sigma$
Initialize: $k \leftarrow 0, t \leftarrow 0, S \leftarrow 0$
**while** $S = 0$ **do**                                           ▷ Stopping criterion
    Choose $X_{t+1} = \cdots = X_{t+m} = k/K$ and observe rewards $Z_{t+1}, \ldots, Z_{t+m}$
    Calculate the confidence bounds for arm $k/K$

$$UB_k \leftarrow \frac{1}{m} \sum_{i=t+1}^{t+m} Z_i + \sigma \sqrt{\frac{2 \log m}{m}}$$

$$LB_k \leftarrow \frac{1}{m} \sum_{i=t+1}^{t+m} Z_i - \sigma \sqrt{\frac{2 \log m}{m}}$$

   **if** $UB_k < LB_i$ for some $i < k$ **then**
      $S \leftarrow 1$
   **else**
      $t \leftarrow t + m, k \leftarrow k + 1$
   **end if**
**end while**
Set $X_t \equiv k/K$ for the remaining $t$

---

bound of the current arm $k/K$ with the lower confidence bound of the historically best arm. If the upper confidence bound of $k/K$ is lower, then it means $k/K$ is likely to have passed the "mode" of $f(x)$. At this point, the algorithm stops escalating and keeps pulling $k/K$ for the rest of the horizon. Clearly, Algorithm 1 satisfies Requirement 1. The regret of Algorithm 1 is provided in Theorem 2.

**Theorem 2.** *By choosing $K = \lfloor T^{1/4} \rfloor$ and $m = \lfloor T^{1/2} \rfloor$, the regret of Algorithm 1 satisfies*

$$R_\pi(T) \leq \left( 3c + \frac{11}{2} + 4\sqrt{3}\sigma \sqrt{\log T} \right) T^{3/4}$$

*when $T \geq 16$, under Assumptions 1 to 3.*

We note that to achieve the regret in Theorem 2, the algorithm requires the information of $T$ and $\sigma$ but not the Lipschitz constant $c$. The requirement for $\sigma$ may be relaxed using data-dependent confidence intervals such as bootstrap, proposed in [28, 23]. The requirement for $T$ is typically relaxed using the well-known doubling trick. However, it may not work in this case because of Requirement 1: once $T$ is doubled, the arm cannot start from $X_t = 0$ again. It remains an open question whether Algorithm 1 can be extended to the case of unknown $T$.

Next we introduce the major steps of the proof of Theorem 2. The discretization error of pulling arms $k/K$, $k \in \{0, \ldots, K\}$, is of the order $O(T/K)$. Therefore, choosing $K = O(T^{1/4})$ controls the regret at the desired rate. The total number of pulls "wasted" in this stage is $O(mK) = O(T^{3/4})$. We consider the good event, on which $f(k/K)$ is inside the confidence interval $[LB_k, UB_k]$ for all $k = 0, 1, \ldots$ until the stopping criterion is triggered. We choose $m$ such that for a given $k$ this happens with a probability no less than $1 - O(1/m)$. Therefore, applying the union bound to at most $K$ discretized arms, the probability of the good event is no less than $1 - O(K/m) = 1 - O(T^{-1/4})$. As a result, the regret incurred due to the bad event is bounded by $TO(T^{-1/4}) = O(T^{3/4})$, matching the desired rate. On the good event, the stopping criterion is triggered because arm $k/K$ has passed $\arg\max_{x \in [0,1]} f(x)$. But it is not much worse than the optimal arm because otherwise the stopping criterion would have been triggered earlier. The gap between $f(k/K)$ and $f(x^*)$ is controlled by the width of the confidence interval, $O(\sqrt{\log m/m})$. Multiplying it by $T$ (because $k/K$ is pulled for the rest of the horizon once the stopping criterion is triggered) gives $O(T\sqrt{\log m/m}) = O(T^{3/4}\sqrt{\log T})$. Putting the pieces together gives the regret bound in Theorem 2.

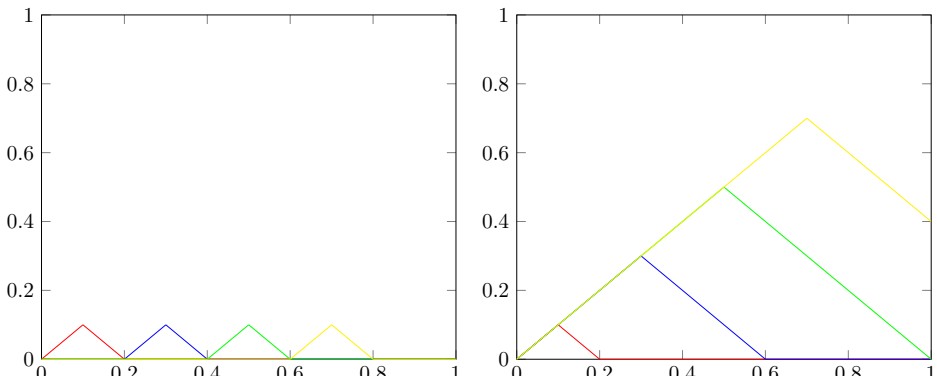

Figure 2: The constructed instances for the lower bound in continuum-armed bandit (left panel) and Theorem 3 (right panel).

## 4 Lower Bound for the Regret

In the classic continuum-armed bandit problem, without Requirement 1, it is well known that there exist policies and algorithms that can achieve regret $O(T^{2/3})$, which has also been shown to be tight [26, 6, 38]. In this section, we show that in the presence of Requirement 1, no policies can achieve regret less than $\Omega(T^{3/4})$ under Assumptions 1, 2 and 3.

**Theorem 3** (Lower Bound). *When $T \geq 16$, under Assumption 1 with $c = 1$, Assumptions 2 and 3, we have*

$$R_\pi(T) \geq \frac{1}{32}T^{3/4},$$

*for any policy $\pi$ satisfying Requirement 1.*

Compared to the literature, Theorem 3 implies that Requirement 1 substantially impedes the learning efficiency by increasing the rate of regret from $T^{2/3}$ to $T^{3/4}$. The proof for Theorem 3 is significantly different from that in the literature. In particular, a typical construction for the lower bound in the continuum-armed bandit literature involves a class of functions that have "humps" in a small interval and are zero elsewhere, as illustrated in the left panel of Figure 2. The idea is to make the functions indistinguishable except for two small intervals. In our case, it is not sufficient to achieve $O(T^{3/4})$. To utilize Requirement 1, the constructed functions in the right panel of Figure 2 share the same pattern before reaching the mode. Afterwards, the functions diverge. The objective is to make it hard for the decision maker to decide whether to keep escalating the arm or stop, and punish her severely if a wrong decision is made. (Consider stopping at the mode of the red function while the actual function is yellow.) Such regret is much larger than the hump functions and thus gives rise to the higher rate.

Besides the construction of the instances, the proof relies on the following random time

$$\tau = \max\{t | X_t \leq a\}$$

for some constant $a$. Note that $\tau$ is typically not a stopping time for a stochastic process $X_t$. Under Requirement 1, however, it is a stopping time. This is the key observation that helps us to fully utilize the monotonicity. Technically, this allows us to study the Kullback-Leibler divergence for the probability measures on $(X_1, Z_1, \ldots, X_\tau, Z_\tau, \tau)$. By decomposing the KL-divergence on the events $\{\tau = 1\}$, $\{\tau = 2\}$, and so on, we can focus on the periods when $X_t \leq a$, which cannot possibly distinguish functions whose mode are larger than $a$ (right panel of Figure 2). This observation and innovation is essential in the proof of Theorem 3.

## 5 Numerical Experiment

In this section, we conduct numerical studies to compare the regret of Algorithm 1 with a few benchmarks. The experiments are conducted on a Windows 10 desktop with Intel i9-10900K CPU.

To make a fair comparison, we generate the instances randomly so that it doesn't bias toward our algorithm. In particular, we consider $T \in \{1000, 11000, 21000, \ldots, 101000\}$. For each $T$, we simulate 100 independent instances. For each instance, we generate $x_1 \sim U(0, 1)$ and $x_2 \sim U(0.5, 1)$, and let $f(x)$ be the linear interpolation of the three points $(0, 0)$, $(x_1, x_2)$, and $(1, 0)$. The reward $Z_t \sim \mathcal{N}(f(X_t), 0.1)$ for all instances and $T$.[1] We consider the following algorithms for each instance:

1. Algorithm 1. As stated in Theorem 2, we use $K = \lfloor T^{1/4} \rfloor$, $m = \lfloor T^{1/2} \rfloor$ and $\sigma = 0.1$.

2. The classic UCB algorithm for the continuum-armed bandit. We first discretize the interval $[0, 1]$ into $K + 1$ equally spaced grid points, $\{k/K\}_{k=0}^{K}$, where $K = \lfloor T^{1/3} \rfloor$ according to [26]. Then we implement the version of UCB in Chapter 8 of [29] on the multi-armed bandit problem with $K + 1$ arms. More precisely, we have

$$X_t = \arg\max_{k/K} (UCB_k(t))$$

$$UCB_k(t) = \begin{cases} 1 & T_k(t - 1) = 0 \\ \hat{\mu}_k(t - 1) + \sigma\sqrt{\frac{2\log(1 + t\log(t)^2)}{T_k(t-1)}} & T_k(t - 1) \geq 1 \end{cases}$$

   where $\mu_k(t - 1)$ is the empirical average of arm $k$, $x = k/K$, in the first $t - 1$ periods; $T_k(t - 1)$ is the number that arm $k$ is chosen in the first $T - 1$ periods. Note that in this benchmark we do not impose Requirement 1.

3. UCB with Requirement 1. We follow the same discretization scheme and UCB definition in the last benchmark, except that we impose $X_{t+1} \geq X_t$. That is

$$X_t = \max\{X_{t-1}, \arg\max_{k/K} (UCB_k(t))\}.$$

4. Deflating UCB with Requirement 1. To improve the performance of UCB with Requirement 1, we need to make the algorithm more likely to stick to the current arm. Moreover, when exploring new arms, we need to encourage the algorithm to explore the arm slightly larger than the current arm. We design a heuristic for this purpose by deflating the UCB initialization of the arms. More precisely, we let

$$UCB_k(t) = 1 - k/K, \text{ if } T_k(t - 1) = 0.$$

   In this way, the decision maker is encouraged to explore arms with small $k$ first and gradually escalate the arms.

The average regret over the 100 instances is shown in Figure 3. We may draw the following observations from the experiment. First, the regret of UCB is significantly less than the other three algorithms which are constrained by Requirement 1. Therefore, the cost brought to the learning efficiency by Requirement 1 is huge. Second, simply imposing Requirement 1 to the UCB algorithm doesn't perform well (the green curve). It more than triples the regret of Algorithm 1. Third, the deflating UCB heuristic seems to perform well. Its regret is on par with that of Algorithm 1, although the trend implies a fast scaling with $T$. The regret analysis and the optimal choice of the deflating factor remain an open question. Overall, Algorithm 1 performs the best among the algorithms that are constrained by Requirement 1. The result is consistent with the theoretical studies.

## 6 Conclusion

In this study, we investigate the online learning or multi-armed bandit problem, when the chosen actions are ordinal and required to be monotone over time. The requirement appears naturally in several applications: firms setting markup pricing policies in order not to antagonize early adopters and clinical trials following dose escalation principles. It turns out that the requirement may severely limit the capability of the decision maker to learn the objective function: the worst-case regret is linear in the length of the learning horizon, when the objective function is just continuous. However, we show that if the objective function is also quasiconcave, then the proposed algorithm can achieve

---

[1]Here $U(a, b)$ denotes a uniform random variable between in $[a, b]$; $\mathcal{N}(a, b)$ denotes a normal random variable with mean $a$ and standard deviation $b$.

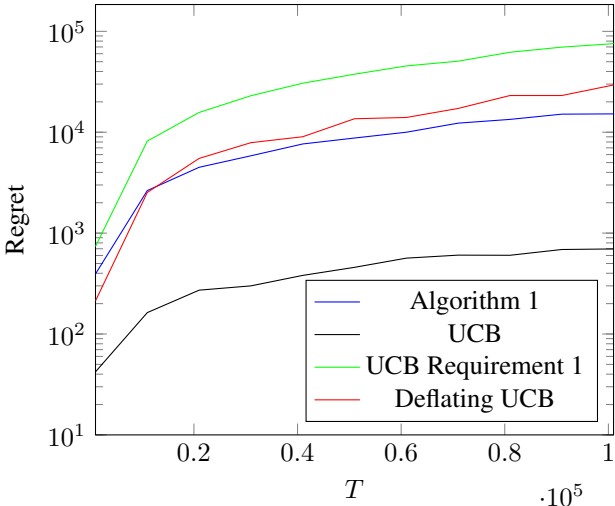

Figure 3: The average regret of Algorithm 1 and three benchmarks.

sublinear regret $\tilde{O}(T^{3/4})$. It is still worse than the optimal rate in the continuum-armed bandit literature, but is proved to be optimal in this setting.

The study has a few limitations and opens up potential research questions. First, since the doubling trick no longer works, the assumption of knowing $T$ in advance cannot be simply relaxed. It is not clear if one can design a new algorithm that can achieve the optimal regret even when $T$ is unknown. Second, it is unclear if the monotone requirement can be incorporated in contextual bandits. This is a highly relevant question in clinical trials, which increasingly rely on patient characteristics for the dose allocation. A novel framework is needed to study how the dose escalation principle plays out in this setting.

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
