# A   Proofs

*Proof of Theorem 2:* Let $x^* \in \arg\max_{x \in [0,1]} f(x)$. Because $K = \lfloor T^{1/4} \rfloor$, we can find $k^*$ such that $|x^* - k/K| \le 1/(2\lfloor T^{1/4} \rfloor)$. Therefore, we have

$$R_\pi(T) = Tf(x^*) - \sum_{t=1}^{T} \mathrm{E}[f(X_t)]$$

$$= Tf(x^*) - Tf(k^*/K) + (Tf(k^*/K) - \sum_{t=1}^{T} \mathrm{E}[f(X_t)])$$

$$\le Tc|x^* - k/K| + (Tf(k^*/K) - \sum_{t=1}^{T} \mathrm{E}[f(X_t)])$$

$$= \frac{cT}{2K} + (Tf(k^*/K) - \sum_{t=1}^{T} \mathrm{E}[f(X_t)]). \tag{1}$$

Here the first inequality is due to Assumption 1. Next we can focus on $Tf(k^*/K) - \sum_{t=1}^{T} \mathrm{E}[f(X_t)])$, the regret relative to the best arm in $\{0, 1/K, \ldots, (K-1)/K, 1\}$.

Consider $m$ IID random rewards $Z'_{k,1}, \ldots, Z'_{k,m}$ having the same distribution as $Z_t$ when $X_t = k/K$, for $k = 0, \ldots, K$. Let $\bar{Z}'_k = \frac{1}{m} \sum_{t=1}^{m} Z'_{k,m}$. Consider the event $E$ as

$$E := \left\{ \bar{Z}'_k - \sigma\sqrt{\frac{2\log m}{m}} \le f(k/K) \le \bar{Z}'_k + \sigma\sqrt{\frac{2\log m}{m}}, \ \forall k \in \{0, \ldots, K\} \right\}.$$

If we couple $Z'_{k,i}$, $i = 1, \ldots, m$, with the rewards generated in the algorithm pulling arm $k/K$, then the event represents the high-probability event that the $f(k/K)$ is inside the confidence interval $[LB_k, UB_k]$. Using the standard concentration bounds for subgaussian random variables (note that $\bar{Z}'_k$ is $\sigma/\sqrt{m}$-suggaussian), we have

$$\mathsf{P}(E^c) \le \cup_{k=0}^{K} \mathsf{P}\left( |\bar{Z}'_k - \mathrm{E}[\bar{Z}'_k]| > \sigma\sqrt{\frac{2\log m}{m}} \right)$$

$$\le (K+1)2\exp\left\{ -\frac{m}{2\sigma^2} \times \frac{2\sigma^2 \log m}{m} \right\} = \frac{2(K+1)}{m}. \tag{2}$$

Based on the event $E$, we can decompose the regret as

$$Tf(k^*/K) - \sum_{t=1}^{T} \mathrm{E}[f(X_t)] = \sum_{t=1}^{T} \mathrm{E}\left[(f(k^*/K) - f(X_t))\mathbb{1}_E\right] + \sum_{t=1}^{T} \mathrm{E}\left[(f(k^*/K) - f(X_t))\mathbb{1}_{E^c}\right]$$

$$\le \sum_{t=1}^{T} \mathrm{E}\left[(f(k^*/K) - f(X_t))\mathbb{1}_E\right] + T\mathsf{P}(E^c)$$

$$\le \sum_{t=1}^{T} \mathrm{E}\left[(f(k^*/K) - f(X_t))\mathbb{1}_E\right] + \frac{2T(K+1)}{m}, \tag{3}$$

where the first inequality follows from $f(k^*/K) \le 1$ and the second inequality follows from (2).

Next we analyze the first term of (3). Suppose $T_1$ is the stopping time when the stopping criterion $S \to 1$ is triggered in Algorithm 1. We can divide the horizon into two phases $[0, T_1]$ and $[T_1 + 1, T]$. Before the stopping criterion, the first term of (3) is bounded by

$$\mathrm{E}\left[ \sum_{t=1}^{T_1} \mathrm{E}\left[(f(k^*/K) - f(X_t))\mathbb{1}_E\right] \right] \le (K+1)mf(k^*/K) \le (K+1)m \tag{4}$$

To analyze the second phase, since we can couple the random variables $Z'_{k,m}$ and the rewards of arm $k/K$, we can suppose that $LB_k \le f(k/K) \le UB_k$ on event $E$ for all $k$ during Algorithm 1. Note that when the stopping criterion $S \leftarrow 1$ is triggered for some arm $k/K$ in Algorithm 1, we must have

$$f(k/K) \le UB_k < LB_i \le f(i/K) \tag{5}$$

for some $i < k$. Note that (5), combined with Assumption 3, implies that $x^* \leq k/K$. Otherwise we have $f(x^*) \geq f(i/K) > f(k/K)$ while $i/K < k/K < x^*$, which contradicts Assumption 3. This fact then implies that $k^* \leq k$ and furthermore $k^* \leq k - 1$ because $f(k/K) \leq f(i/K)$.

Because the stopping criterion is triggered for the first time, it implies that

$$f((k-1)/K) \geq LB_{k-1} = UB_{k-1} - 2\sigma\sqrt{\frac{2\log m}{m}} \geq LB_i - 2\sigma\sqrt{\frac{2\log m}{m}}$$

$$\geq UB_i - 4\sigma\sqrt{\frac{2\log m}{m}} \geq f(i/K) - 4\sigma\sqrt{\frac{2\log m}{m}}. \tag{6}$$

Here the first inequality is due to event $E$. The first equality is due to the definition of $UB$ and $LB$. The second inequality is due to the fact that the stopping criterion is not triggered for arm $k'/K$. The last inequality is again due to event $E$. Moreover, because arm $i/K$ is historically the best among $\{0, 1/K, \dots, (k-1)/K\}$, we have

$$f(i/K) \geq LB_i \geq LB_{k'} = UB_{k'} - 2\sigma\sqrt{\frac{2\log m}{m}} \geq f(k'/K) - 2\sigma\sqrt{\frac{2\log m}{m}} \tag{7}$$

for all $0 \leq k' \leq k - 1$. Now (6) and (7), combined with $k^* \leq k - 1$, imply that

$$f\left(\frac{k-1}{K}\right) \geq f(i/K) - 4\sigma\sqrt{\frac{2\log m}{m}} \geq f(k^*/K) - 6\sigma\sqrt{\frac{2\log m}{m}}.$$

By Assumption 1, we then have

$$f(k/K) \geq f\left(\frac{k-1}{K}\right) - \frac{c}{K} \geq f(k^*/K) - \frac{c}{K} - 6\sigma\sqrt{\frac{2\log m}{m}}.$$

Plugging the last inequality back into the first term of (3) in the second phase, we have

$$\mathrm{E}\left[\sum_{t=T_1+1}^{T} \mathrm{E}\left[(f(k^*/K) - f(X_t))\mathbb{1}_E\right]\right] \leq T(f(k^*/K) - f(k/K)) \leq \frac{cT}{K} + 6\sigma\sqrt{\frac{2\log m}{m}}T. \tag{8}$$

Combining (1), (3), (4) and (8), we have

$$R_\pi(T) \leq \frac{cT}{2K} + \frac{2(K+1)T}{m} + (K+1)m + \frac{cT}{K} + 6\sigma\sqrt{\frac{2\log m}{m}}T$$

$$\leq 3cT^{3/4} + 4T^{3/4} + \frac{3}{2}T^{3/4} + 4\sqrt{3}\sigma\sqrt{\log T}T^{3/4}$$

$$\leq \left(3c + \frac{11}{2} + 4\sqrt{3}\sigma\sqrt{\log T}\right)T^{3/4},$$

where we have plugged in $K = \lfloor T^{1/4} \rfloor$ and $m = \lfloor T^{1/2} \rfloor$, and moreover,

$$K \leq T^{1/4} \leq 2K, \ K + 1 \leq \frac{3}{2}T^{1/4}, \ \frac{3}{4}T^{1/2} \leq T^{1/2} - 1 \leq m \leq T^{1/2},$$

because $T \geq 16$. This completes the proof.

$\square$

*Proof of Theorem 3:* Let $K = \lfloor T^{1/4} \rfloor$ and construct a family of functions $f_k(x)$ as follows. For $k \in [K]$, let

$$f_k(x) = \begin{cases} x & x \in [0, (k-1/2)/K) \\ \max\{(2k-1)/K - x, 0\} & x \in [(k-1/2)/K, 1] \end{cases}$$

As a result, we can see that $\max_{x \in [0,1]} f_k(x) = (k-1/2)/K$ is attained at $x = (k-1/2)/K$. Clearly, all the functions satisfy Assumption 1 with $c = 1$ and Assumption 3. For each $f_k(x)$, we construct the associated reward sequence by $Z_t \sim \mathcal{N}(f_k(X_t), 1)$, which is a normal random variable with mean $f_k(X_t)$ and standard deviation 1. It clearly satisfies Assumption 2.

Consider a particular policy $\pi$. Let
$$R_k := R_{f_k, \pi}(T)$$
be the regret incurred when the objective function is $f_k(x)$ for $k \in [K]$. Because of the construction, it is easy to see that for the objective function $f_k(x)$, if $X_t \notin [(k-1)/K, k/K]$, then a regret no less than $1/(2K)$ is incurred in period $t$. Therefore, we have

$$R_k \geq \frac{1}{2K} \sum_{t=1}^{T} \mathrm{E}_k[\mathbb{1}_{X_t \notin [(k-1)/K, k/K]}]. \tag{9}$$

Here we use $\mathrm{E}_k$ to denote the expectation taken when the objective function is $f_k(x)$. On the other hand, if we focus on $R_K$, then it is easy to see that

$$R_K \geq \left(\frac{1}{2} - \frac{1}{2K}\right) \sum_{t=1}^{T} \mathrm{E}_K[\mathbb{1}_{X_t \leq \lfloor K/2 \rfloor / K}], \tag{10}$$

because a regret no less than $1/2 - 1/2K$ is incurred in the periods when $X_t \leq 1/2$.

Based on the regret decomposition in (9) and (10), we introduce $T_{k,i}$ for $k, i \in [K]$ as

$$T_{k,i} = \sum_{t=1}^{T} \mathrm{E}_k[\mathbb{1}_{X_t \in [(i-1)/K, i/K)}].$$

In other words, $T_{k,i}$ is the number of periods in which the policy chooses $x$ from the interval $[(i-1)/K, i/K)$ when the reward sequence is generated by the objective function $f_k(x)$.[2] A key observation due to Requirement 1 is that

$$T_{i+1,i} = T_{i+2,i} = \cdots = T_{K,i}. \tag{11}$$

This is because for $k > i$, the function $f_k(x)$ is identical for $x \leq i/K$. Before reaching some $t$ such that $X_t > i/K$, the policy must have spent the same number of periods on average in the interval $[(i-1)/K, i/K)$ no matter the objective function is $f_{i+1}(x), \ldots, f_{K-1}(x)$, or $f_K(x)$. But because of Requirement 1, once $X_t > i/K$ for some $t$, the policy never pulls an arm in the interval $[(i-1)/K, i/K)$ afterwards. Therefore, (11) holds. This allows us to simplify the notation by letting $T_i := T_{k,i}$ for $k > i$. In particular, by (10), we have

$$R_K \geq \frac{K-1}{2K} \sum_{i=1}^{\lfloor K/2 \rfloor} T_{K,i} = \frac{K-1}{2K} \sum_{i=1}^{\lfloor K/2 \rfloor} T_i. \tag{12}$$

Next we are going to show the relationship between $T_{k,i}$ (or equivalently $T_i$) and $T_{i,i}$ for $k > i$. We introduce a random variable $\tau_i$
$$\tau_i := \max\{t | X_t < i/K\}.$$
Because of Requirement 1, we have $\{\tau_i \leq t\} \in \sigma(X_1, Z_1, X_2, Z_2, \ldots, X_t, Z_t, U_t)$.[3] Therefore, $\tau_i$ is a stopping time. We consider the two probability measures, induced by the objective functions $f_i(x)$ and $f_k(x)$ respectively, on $(X_1, Z_1, \ldots, X_{\tau_i}, Z_{\tau_i}, \tau_i)$. Denote the two measures by $\mu_{i,i}$ and $\mu_{k,i}$ respectively. Therefore, we have

$$T_{i,i} - T_{k,i} = \left( \mathrm{E}_{\mu_{i,i}} \left[ \sum_{t=1}^{\tau_i} \mathbb{1}_{X_t \in [(i-1)/K, i/K)} \right] - \mathrm{E}_{\mu_{k,i}} \left[ \sum_{t=1}^{\tau_i} \mathbb{1}_{X_t \in [(i-1)/K, i/K)} \right] \right)$$
$$\leq T \sup_A (\mu_{i,i}(A) - \mu_{k,i}(A)) \tag{13}$$
$$\leq T \sqrt{\frac{1}{2} D(\mu_{i,i} \parallel \mu_{k,i})}. \tag{14}$$

---

[2] We let $T_{k,K} = \sum_{t=1}^{T} \mathrm{E}_k[\mathbb{1}_{X_t \in [1-1/K, 1]}]$ include the right end. This is a minor technical point that doesn't affect the steps of the proof.

[3] Recall that $U_t$ is an internal randomizer. Since we can always couple the values of $U_t$ under the two measures, we omit the dependence hereafter.

Here (13) follows the definition of the total variation distance and the fact that $\sum_{t=1}^{\tau_i} \mathbb{1}_{X_t \in [(i-1)/K, i/K)} \leq T$. The second inequality (14) follows from Pinsker's inequality (see [40] for an introduction) and $D(P \parallel Q)$ denotes the Kullback-Leibler divergence defined as

$$D(P \parallel Q) = \int \log(\frac{dP}{dQ})dP.$$

We can further bound the KL-divergence in (14) by:

$$
\begin{aligned}
D(\mu_{i,i} \parallel \mu_{k,i}) &= \sum_{t=1}^{T} \int_{\tau_i=t} \log\left(\frac{\mu_{i,i}(x_1, z_1, \ldots, x_t, z_t)}{\mu_{k,i}(x_1, z_1, \ldots, x_t, z_t)}\right) d\mu_{i,i} \\
&= \sum_{t=1}^{T} \int_{\tau_i=t} \sum_{s=1}^{t} \log\left(\frac{\mu_{i,i}(z_s|x_s)}{\mu_{k,i}(z_s|x_s)}\right) d\mu_{i,i} \\
&= \sum_{t=1}^{T} \int_{\tau_i=t} \int_{z_1,\ldots,z_t} \sum_{s=1}^{t} \log\left(\frac{\mu_{i,i}(z_s|x_s)}{\mu_{k,i}(z_s|x_s)}\right) d\mu_{i,i}(z_1, \ldots, z_s|x_1, \ldots, x_t) d\mu_{i,i}(x_1, \ldots, x_t) \\
&= \sum_{t=1}^{T} \int_{\tau_i=t} \int_{z_1,\ldots,z_t} \sum_{s=1}^{t} D(\mathcal{N}(f_i(x_s), 1) \parallel \mathcal{N}(f_k(x_s), 1)) d\mu_{i,i}(x_1, \ldots, x_t) \\
&= \sum_{t=1}^{T} \int_{\tau_i=t} \sum_{s=1}^{t} \frac{1}{2}(f_i(x_s) - f_k(x_s))^2 d\mu_{i,i}(x_1, \ldots, x_t)
\end{aligned}
$$

In the first line we use the fact that the normal reward has support $\mathbb{R}$. Hence if the sample path $(x_1, z_1, \ldots, x_t, z_t)$ has positve density under $\mu_{i,i}$ then it has positive density under $\mu_{k,i}$. As a result, we establish the absolute continuity of $\mu_{i,i}$ w.r.t. $\mu_{k,i}$ and the existence of the adon-Nikodym derivative. The second equality follows from the fact that for the same policy $\pi$, we have

$$\mu_{i,i}(x_s|x_1, z_1, \ldots, x_{s-1}, z_{s-1}) = \mu_{k,i}(x_s|x_1, z_1, \ldots, x_{s-1}, z_{s-1}).$$

The fourth equality uses the conditional independence of $z$ given $x$. Note that on the event $\tau_i = t$, we have $x_s < i/K$ for $s \leq t$. Therefore, we have

$$|f_i(x_s) - f_k(x_s)| \leq \begin{cases} \frac{1}{K} & x_s \in [(i-1)/K, i/K) \\ 0 & x_s < (i-1)/K \end{cases},$$

by the construction of $f_i(x)$ and $f_k(x)$. As a result,

$$D(\mu_{i,i} \parallel \mu_{k,i}) \leq \sum_{t=1}^{T} \int_{\tau_i=t} \frac{T_{k,i}}{2K^2} d\mu_{i,i}(x_1, \ldots, x_t) \leq \frac{T_{k,i}^2}{2K^2}.$$

Plugging it into (14), we have Therefore, (14) implies

$$T_{i,i} \leq T_{k,i} + \frac{T}{2K}\sqrt{T_{k,i}} = T_i + \frac{T}{2K}\sqrt{T_i}. \tag{15}$$

Combining (15) and (9), we can provide a lower bound for the regret $R_i$ for $i = 1, \ldots, \lfloor K/2 \rfloor/K$:

$$R_i \geq \frac{1}{2K}(T - T_{i,i}) \geq \frac{1}{2K}\left(T - T_i - \frac{T}{2K}\sqrt{T_i}\right). \tag{16}$$

Next, based on (12) and (16), we show that for $k \in \{1, \ldots, \lfloor K/2 \rfloor/K, K\}$, there exists at least one $k$ such that

$$R_k \geq \frac{1}{32}T^{3/4}.$$

If the claim doesn't hold, then we have $R_K \geq T^{3/4}/32$. By (12) and the pigeonhole principle, there exists at least one $i$ such that

$$T_i \leq \frac{2K}{32(K-1)\lfloor K/2 \rfloor}T^{3/4}.$$

Because $T \geq 16$ and $K \geq 2$, we have $K/(K-1) \leq 2$ and $\lfloor K/2 \rfloor \geq T^{1/4}/4$. Therefore,

$$T_i \leq \frac{1}{2} T^{1/2}.$$

Now by (16), for this particular $i$, we have

$$R_i \geq \frac{1}{2K} \left( T - \frac{1}{2} T^{1/2} - \frac{T}{2K} \sqrt{T^{1/2}/2} \right)$$

$$\geq 2T^{-1/4} \left( T - \frac{1}{2} T^{1/2} - \frac{\sqrt{2}}{2} T \right) \tag{17}$$

$$\geq \left( \frac{7}{4} - \sqrt{2} \right) T^{3/4} \geq \frac{1}{32} T^{3/4}, \tag{18}$$

resulting in a contradiction. Here (17) follows from the fact that $2K \geq T^{1/4} \geq K$ when $T \geq 16$; (18) follows from $T^{1/2} \geq 4$. Therefore, we have proved that for at least one $k$, $R_k \geq T^{3/4}/32$. This completes the proof. $\qquad \square$