# OpenReview forum: "Multi-armed Bandit Requiring Monotone Arm Sequences"
_NeurIPS.cc/2021/Conference — NeurIPS 2021 Poster_

### Official Review · Reviewer_YioK · 2021-06-24

**Rating:** 7
**Confidence:** 5

**Summary:**

This paper studies the continuous MAB problem under the constraint that the sequence (X_t)_t of chosen arms is non-decreasing. Obtaining non-linear regrets is obviously impossible without restricting the class of candidate functions: the authors focus on the quasi-concave (or unimodal) Lipschitz case.
They show that the monotonicity constraint on the choices of arms causes the regret to grow as O(T^3/4) instead of O(T^2/3) for the Lipschitz case and O(T^1/2) for the quasi-concave case. A rather straightforward algorithm reaches this rate up to logarithmic factors. A T^{3/4}/32 lower bound is proved.



**Ethical Concerns:**

nothing to say

**Limitations And Societal Impact:**


no applicable here

**Main Review:**

Globally, the paper is pleasant to read. The problem solved is rather simple but the treatment is nice and clean.

The motivation for the problem is somewhat questionable: in clinical trials, it is possible to lower the dose if too high toxicity is detected, and in dynamic pricing experience shows that the sequence of prices is rarely monotonous (even though we would prefer if it were). The justification of this setting in the introduction sounds somewhat naive, and is repeated pretty redundantly in Section 2.

Proposition 1 (the impossibility to have non-linear regrets in general) is pretty obvious, but nicely written. In general, the ideas come very naturally, and are well exposed. There are a number of shortcomings listed below, which absolutely need to be addressed. The main (reasonably simple) thing missing in my view is a lower bound showing the impossibility to have a non-linear regret if the horizon T is unknown (this is cited as an open question after Thm.2). I would suggest to change the name of the paper, which I find slightly misleading, to "escalating bandits".

But apart from that, I think that the paper has a nice message and seems suited to a publication at a conference.


Important things to correct :

- the sketch of proof after Thm.2 is nice and gives most ideas, except one term which seems to be forgotten: the contribution to the regret of the O(mK) draws to the initial arms before the maximum is reached. This term explains why m should not be chosen to be too large: it is important for the intuition. I hence think that this should be corrected.

- in the supplementary material on line 453, you should recall here what is T_1

- in the supplementary material, there are several problems in the proof of Theorem 3 that need to be fixed:

  * the sequence of equations after line 506 is not rigorously written: "\mu_{i,i}(x_1,z_1,\dots,x_t,z_t)" is zero everywhere. But since \mu_{i,i} is absolutely continuous with respect to \mu{k,i} (to be justified), the Radon-Nikodym derivative d\mu_{i,i}/d\mu_{k,i} exists. On the third line, it should not be d\mu_{i,i}(\cdot |...) but d\mu_{i,i}(z_1,\dots,z_s |...), and you could use the conditional independence to simplify the expression here. Globally, I think that this entire sequence of equations needs to be rewritten (I know that this type of Lai-Robbins lower bound is a pain to write, but more clarity is still needed).

  *  in the display of line 509, the T_{k,i} term should not be squared.




**Time Spent Reviewing:**

5

---

> ### Author Response · Authors · 2021-08-10
> **Response to Reviewer YioK**
>
> Thank you for your insightful and encouraging comments.
>
> - *The motivating examples are questionable.*
>
>   Thank you for raising this point. Since your point is similar to the one given by P9sB, we are pasting our response here. Below we provide more details on the applications and argue why monotonicity constraints are important.
>
>   1. Dynamic pricing with markdown. Markdown pricing is common in retailing. In fact, most retail stores do not increase the price of a seasonal or perishable product despite the fact that the product is being sold successfully. A recent survey by Google [3] also mentions the importance of markdown optimization in the retail sector. Group buying is another example in which the seller lowers the price gradually with more sign-ups [4].
>   2. Dynamic pricing with markup. In successful crowdfunding campaigns, markup pricing is often used:  to reward early backers, the initial products may be given for free; later after some goals are achieved, backers may get the product at a discount; after the release, the product is sold at the full price. It is not a good idea to antagonize early backers by lowering the price later on. For digital services growing rapidly, markup pricing is also a typical strategy: both Dropbox and Google Drive went through the phases from free, to low fees, to high fees.
>   3. Clinical trials. Early-stage trials are conducted in Phase I and II sequentially. For Phase I, the toxicity of the doses is examined to determine a safe dose range. In this phase, dose escalation is imposed and "3+3 design" is the most common dose allocation approach. For Phase II, the efficacy of the doses are tested within the safe dose range (by random assignment, for example). Recently, especially in oncology, there are proposals to integrate both phases [5] which is called "seamless Phase I/II trials." In this case, dose escalation has to be observed when looking for the most efficacious dose. Our study is thus motivated by seamless trials.
>
>   We agree with you that the monotonicity is not necessarily a hard constraint in the examples. However, violating the monotonicity usually introduces a cost that needs external modeling to quantify, such as the backlash of consumers in dynamic pricing or ethical issues in clinical trials. Therefore, we impose it as a hard constraint for a parsimonious model.
>
> - *Linear regret for unknown T.*
>
>   This is a great point. We agree with your intuition that a lower bound of $\Omega(T)$ can be established. Here is a basic argument for policies without internal randomization. We can consider the class of reward functions with $f(0)=0$. For any policy, let $\tau = \min_t\{t:X_t>0\}$ be the first time that $X_t$ is strictly positive. Because $T$ is unknown and $f(0)=0$,  $\tau$ and $X_\tau$ must be chosen independent of $f(x)$ and $T$. One can always find $f(x)$ whose maximum is less than $X_\tau/2$. For such a reward function, from $\tau$ on, the per-period regret of the policy is at least $f(X_{\tau/2})-f(X_\tau)$ and not diminishing as $T$ increases. This shows that the regret would grow linearly. We believe the argument works for policies with internal randomization as well by considering $(\tau, X_\tau)$​​​ as random variables independent of everything else.
>
> - *Change of the title to "escalating bandits"*
>
>   Thank you for this suggestion. If the review team believes "Escalating Bandits" is a better title. We would be willing to change it.
>
> - *The $O(mK)$ term in the regret.*
>
>   Thank you for catching this. We will add this term in the revision.
>
> - *Recall $T_1$ in 453.*
>
>   Thank you for the careful reading. We will clarify the notation.
>
> - *Problems in the proof of Theorem 3.*
>
>   Thank you so much for catching these. We will fix them in the revision.
>
> [3] Google (2021). Transforming specialty retail with AI. Technical report.
>
> [4] Hu, M., Liu, J., & Zhai, X. (2020). Intertemporal segmentation via flexible-duration group buying. *Manufacturing & Service Operations Management*.
>
> [5] Wages, N. A., & Tait, C. (2015). Seamless phase I/II adaptive design for oncology trials of molecularly targeted agents. *Journal of biopharmaceutical statistics*, *25*(5), 903-920.

---

> > ### Comment · Reviewer_YioK · 2021-08-27
> > **Proof of Theorem 3.**
> >
> > Dear authors,
> > regarding the proof of Theorem 3, could you please detail a little how to plan to fix the problems in the final version?
> > Best regards.

---

> > > ### Author Response · Authors · 2021-08-27
> > > **Plan to fix Theorem 3**
> > >
> > > Dear reviewer,
> > >
> > > We will follow your suggestion to streamline and tighten up the sequence of inequalities after line 506. In particular, for the first line, we will establish that for any arm sequence $(x_1,x_2,...,x_T)$ that can possibly be taken by the policy, we have $\mu_{i,i}>0$ and $\mu_{k,i}>0$. This is because the normal distribution we use for the rewards has support $\mathbb R$ so any $(x_1,z_1,...,x_\tau, z_\tau, \tau)$ has a positive density (given that the arm sequence is compatible with the policy). This will lead to your argument about Radon-Nikodym derivative and absolute continuity.
> > >
> > > For the third line, you pointed out the fix exactly and we will simply follow your comment.
> > >
> > > Regarding your comment of rearranging the sequence of inequalities, we could not come up with a different and better way to establish the bound on the KL-divergence. If you believe a different chain of inequalities would read better, please feel free to let us know.
> > >
> > > For line 509, you are exactly right and we thank you for pointing out the typo. We will fix it in the revision.

---

> > > > ### Comment · Reviewer_YioK · 2021-09-10
> > > > **Re**
> > > >
> > > > Thank you for your answers. Taking everything into consideration, I keep my rating as-is.

---

### Official Review · Reviewer_P9sB · 2021-07-12

**Rating:** 7
**Confidence:** 4

**Summary:**

The paper studies a bandit problem where the actions need to be monotone. Precisely, given an
action space [0, 1], a decision-maker needs to choose actions X_1, ..., X_T, where
X_1 <= X_2 <= ... <= X_T. The authors first show that in the worst case, O(T) regret is unavoidable.
Under a quasi-concavity assumption, they show that T^{3/4} regret is both sufficient and necessary.

Overall, I am ambivalent about the paper. The paper has been written well and while I have not
checked the proofs, the results appear correct in hindsight. However, the setting lacks motivation
and the proof techniques, while not straightforward, are not earth-shattering.
This is a fairly subjective assessment, and therefore, I have given a neutral score.


**Main Review:**

There are many positives in the paper. It is clearly written, easy to read, and the main proof
intuitions have been conveyed clearly and concisely.

My main criticism of the paper is that it lacks motivation.  In both examples, while the monotonicity
constraint is at best a 'nice to have', and certainly not a hard constraint.
In the dynamic pricing example, prices change both upwards/downwards all the time based on demand.
I can also think of examples where the opposite is true. Companies have a high price in the
beginning as a form of price discrimination and then adjust both downwards based on the demand.
But there is rarely a hard monotonicity constraint. In the clinical trials example, based on google
search, it seems like dose escalation usually refers to the practice of finding the threshold for
a treatment with potentially toxic side effects. Its usually not the case, that one is
simultaneously finding an optimum treatment amount.
- If the authors can convince me otherwise, I might be willing to bump up my score. For instance, can
you point to some real world use cases (not theory papers) where a seller has not reduced the
price or a clinical trial where dose escalation was done simultaneously while finding an optimal
treatment.

There are many great theory papers where the use case may not be practically well motivated. However,
in such papers, the proofs require very novel techniques and future papers can build on such ideas.
Unfortunately, I don't feel that this paper meets this bar. While the proofs here do require some
novelty, it fundamentally builds on well known techniques.

Question: How would the results change if you were to allow an epsilon-monotonicity constraint? Say,
you are not allowed to make an action now that is more than epsilon less than the previous highest
action, where epsilon could depend on the Lipschitz constant.

---------------------------------
Post rebuttal: I thank the authors for their response. The authors have convinced me sufficiently of the practical significance of the problem and I encourage them to discuss them in more detail in the paper. I also wish to reiterate that the proof does use some nice ideas. While I am not ready to champion the paper, I think it could be a nice addition to the proceedings. I have updated my score to reflect this.

**Time Spent Reviewing:**

3

---

> ### Author Response · Authors · 2021-08-10
> **Response to Reviewer P9sB**
>
> We thank you for your constructive and insightful comments. Please see our response below.
>
> - *The two examples, dynamic pricing and clinical trials, are not well-motivated.*
>
>   We apologize for not elaborating on the examples in the paper. Below we provide more details on the practices and argue why monotonicity constraints are important in the applications.
>
>   1. Dynamic pricing with markdown. Markdown pricing is common in retailing. In fact, most retail stores do not increase the price of a seasonal or perishable product despite the fact that the product is being sold successfully. A recent survey by Google [3] also mentions the importance of markdown optimization in the retail sector. Group buying is another example in which the seller lowers the price gradually with more sign-ups [4].
>   2. Dynamic pricing with markup. In successful crowdfunding campaigns, markup pricing is often used:  to reward early backers, the initial products may be given for free; later after some goals are achieved, backers may get the product at a discount; after the release, the product is sold at the full price. It is not a good idea to antagonize early backers by lowering the price later on. For digital services growing rapidly, markup pricing is also a typical strategy: both Dropbox and Google Drive went through the phases from free, to low fees, to high fees.
>   3. Clinical trials. Early-stage trials are conducted in Phase I and II sequentially. For Phase I, the toxicity of the doses is examined to determine a safe dose range. In this phase, dose escalation is imposed and "3+3 design" is the most common dose allocation approach. For Phase II, the efficacy of the doses are tested within the safe dose range (by random assignment, for example). Recently, especially in oncology, there are proposals to integrate both phases [5] which is called "seamless Phase I/II trials." In this case, dose escalation has to be observed when looking for the most efficacious dose. Our study is thus motivated by seamless trials.
>
>   We agree with you that the monotonicity is not necessarily a hard constraint in the examples. However, violating the monotonicity usually introduces a cost that needs external modeling to quantify, such as the backlash of consumers in dynamic pricing or ethical issues in clinical trials. Therefore, we impose it as a hard constraint for a parsimonious model.
>
> - *What if we allow for $\epsilon$-monotonicity*?
>
>   That is an interesting direction we haven't thought about. We will certainly explore it more when revising the paper. We have some preliminary thoughts: by our analysis, the algorithm can identify the maximum after passing it by a margin of $T^{-1/4}$​​​​.  Therefore, at this point, if $\epsilon$​​​​ is not scaling with $T$​​​​, then the algorithm can always turn back and approximate the maximum arbitrarily well. In this case, we may hope to reclaim the $O(T^{2/3})$​​​​ regret bound by redesigning the discretization scheme.
>
> [3] Google (2021). Transforming specialty retail with AI. Technical report.
>
> [4] Hu, M., Liu, J., & Zhai, X. (2020). Intertemporal segmentation via flexible-duration group buying. *Manufacturing & Service Operations Management*.
>
> [5] Wages, N. A., & Tait, C. (2015). Seamless phase I/II adaptive design for oncology trials of molecularly targeted agents. *Journal of biopharmaceutical statistics*, *25*(5), 903-920.

---

### Official Review · Reviewer_HdAN · 2021-07-16

**Rating:** 6
**Confidence:** 4

**Summary:**

The paper proposes a continuum-armed bandit variant, where the arm sequence needs to be increasing. For the case of Lipschitz functions a linear lower bound is provided. When the functions are constrained to be quasi-concave, a simple algorithm is shown to achieve T^3/4 regret, which is shown to be log-optimal.


**Ethical Concerns:**

-

**Limitations And Societal Impact:**

Yes

**Main Review:**

While the variant seems interesting, the motivating applications are not exactly good fit for a bandit approach. Allocating the dose in subsequent phases of a clinical trial would involve only a small number of decisions, which is insufficient for a bandit algorithm to use its asymptotic strength. Dynamic pricing (on some products) is more interesting for bandits, but not if it is limited to the initial price setting, where the number of decisions is also more limited. I am not sure, if there are better applications, and for that reason the variant seems more of a theoretical interest rather than practical.

The proposed algorithm is a natural one: estimate at a value, increase the value, and stop the increase, if the maximum is overshot. However, there is not much else that can be done under the monotonic constraint without sufficient smoothness, such as (strong-)concavity.

It is good to see new lower-bound constructions, differing from the standard bump design. The standard KL ideas do come in, but there are some additional ideas as well.

A minor point regarding the regret: multiple times it is mentioned that the monotonic constraint brings the regret from T^2/3 to T^3/4. It is not clear for me if one can achieve T^1/2 (as in Table 1 for [17]) without monotonicity, or one needs more constraint on the function other that quasi-concavity. It would be helpful to clear this confusion in the paper. Obviously, T^2/3 can be achieved with the Lipschitz constraint.

There are some numerical experiments, but they are not very informative. Being a new variant, there are no real baselines. Perhaps, it would have been useful to have a concave test problem for which [33] would provide a string baseline. Plotting the theoretical bounds would be also useful.


**Time Spent Reviewing:**

5

---

> ### Author Response · Authors · 2021-08-10
> **Response to Reviewer HdAN**
>
> We thank you for your constructive and insightful comments. Please see our response below.
>
> - *A small number of actions in the motivating examples.*
>
>   Thank you for this point. We discuss the motivating examples extensively in our response to P9sB. Particularly for your point, we agree that in many realistic examples, the number of actions taken is not sufficient for the asymptotic regime to kick in. Still, the bandit framework provides powerful lens to the problem in a structured way. To give an example in clinical trials, in [2] the authors compare a learning algorithm based on the bandit framework to the traditional "3+3 design" for dose allocation in Phase I trials. In 3+3 design, three patients are allocated a dosage level. If none experience dose-limiting toxicity (DLT), then three more patients are assigned to a higher dose; if one experiences DLT, then three more patients are allocated to the same dose for further testing; if two or three experience DLT, then the trial terminates outputting the safe dose range. As you can see, this is highly heuristic. The authors of [2] are able to show that for less than 100 total patients ($T$​), the designed learning algorithm can outperform the traditional approach. The take-away message from [2] is that even a slightly optimized learning algorithm may improve the existing methods, even when the regret analysis doesn't help much for very small $T$​.
>
> - *The regret $T^{1/2}$ in [17] in Table 1.*
>
>   We will elaborate on this point. Like you correctly pointed out, except for quasiconcavity, [17] does assume certain smoothness conditions around the maximum. See Assumption 2 in their paper. It is likely the combination of both conditions that improves $T^{2/3}$ in Lipschitz bandit to $T^{1/2}$.
>
> - *More informative numerical experiments.*
>
>   We will work on the theoretical benchmark as well as other algorithms that assume concavity for example.
>
> [2] Chen, N., & Khademi, A. (2020). Adaptive Seamless Dose-Finding Trials. Working Paper.

---

### Official Review · Reviewer_BiUb · 2021-07-16

**Rating:** 6
**Confidence:** 4

**Summary:**

The paper studies the continuum-armed bandit problem with additional requirement that the selected arms/actions must be monotone sequences over the course of the game. With this requirement, the authors first show that simply considering a continuous reward function (namely, an unknown function that maps the action to the reward) will suffer linear regrets, in contrast to the sublinear regret $O(T^{2/3})$ in the literature without the monotonicity requirement.
Then the authors consider a more structured reward function which is quasiconcave. The authors then provide an algorithm and prove that a regret $O(T^{3/4})$ of this algorithm for this class of functions. Furthermore, this regret is tight as authors also provide a matching lower bound. Lastly, the authors conduct numerical studies to compare the regret of proposed algorithm with a few benchmark, validating the analysis.


**Limitations And Societal Impact:**

Yes

**Main Review:**

Originality, quality, and significance:

Though motivation behind the problem is well-defined, I’m not sure the setting discussed in this paper is a stand alone contribution as an earlier paper [1] has already studied this setting but with concave reward function and showing a regret with order $O(T^{1/2})$. For this paper, the algorithm, together with the idea, is straightforward, which is nice, and the theoretical analysis seems also to be correct.

Clarity:

The paper is overall well written and easy to follow. The authors probably want to consider to move more content in the appendix to the main text as you have enough space.

Other comments:

1.I’m not following your derivations from 156 -- 157, could the authors provide more explanations here?

2.In your theorem 2, when $\sigma = 0$, meaning that your observation is noiseless, the regret still has the order $O(T^{3/4})$. This seems to suggest a pretty large gap. As when   $\sigma = 0$ and without the monotone sequences action requirement, the problem simply reduces to a binary search, which has a regret of $\log T$. Could the authors briefly comment a little bit on this?

[1]. Jad Salem, Swati Gupta, and Vijay Kamble. Taming wild price fluctuations: Monotone stochastic convex optimization with bandit feedback

===== Post Rebuttal =======
Thanks for the authors' response and clarifications, I increased my score to 6.

**Time Spent Reviewing:**

10

---

> ### Author Response · Authors · 2021-08-10
> **Response to Reviewer BiUb**
>
> We thank you for your constructive and insightful comments. Please see our response below.
>
> - *Concave objectives with monotonicity constraints have been studied.*
>
>   Thanks for this point. We would like to argue that this paper contributes to the literature in both practical and theoretical terms. Practically, concave functions are actually relative rare in most applications. For example, if the reward function is the profit and the arm is the charged price, then the reward function is typically not concave because it will eventually flatten to zero as the price increases. In clinical trials, if the reward function is the efficacy and the arm is the dosage, then for many drugs the efficacy plateaus for high doses [1]. In both cases, quasiconcave functions capture the shape well. Theoretically, the regret bound for concave/quasiconcave functions are different and the algorithms have distinct designs. We hope to add more discussion in the revision to clarify the contribution of the paper.
>
> - *Move appendix to the main text.*
>
>   We will address the comments and add more discussion/proof sketches to the paper.
>
> - *The derivation from 156 -- 157 is not clear.*
>
>   We apologize for the confusion. In 156, when the reward function is $f_1$, the oracle gains total reward $T/2$, and if the arm is between $[0,1/2]$, then the maximum reward is $1/2$ in that period. Taking the difference, the regret of $R_{f_1,\pi}$ is at least $T/2-\sum_t 1_{X_t\leq 0.5}/2$. Similarly, for reward function $f_2$, the oracle gains reward $T$. If the arm is between $[0,1/2]$ ($[1/2,1]$), then the maximum reward is $1/2$ ($1$). This gives the regret $R_{f_2,\pi}$. Summing up $R_{f_1,\pi}$ and $R_{f_2,\pi}$, and using the fact that $\sum_t 1_{X_t\leq 0.5}+\sum_t 1_{X_t> 0.5}=T$, we obtain line 157.  We hope this clarifies your confusion and we will provide more details for the derivation in the revision.
>
> - *What happens when $\sigma=0$*?
>
>   Thank you for this thought-provoking point. For the noiseless case, we may discretize the arm interval $[0,1]$ into $T^{1/2}$ arms. In this case, we may let the policy escalate the arms and sample each arm only once. The discretization error is $T^{-1/2}$ per period and it takes at most $T^{1/2}$ trials to find the maximum. The total regret is going to be $O(T^{1/2})$ and we believe it is also tight, using a similar construction in the lower bound. Therefore, you are right that our upper/lower bounds only work for a constant $\sigma>0$.
>
>   The new regret bound is better than Theorem 2, but not as good as $\log(T)$​​ in binary search. (Note that in binary search the reward function is monotone but the search sequence is not.) We believe the discrepancy mainly arises due to the monotonicity requirement.
>
> [1] Riviere, M. K., Yuan, Y., Jourdan, J. H., Dubois, F., & Zohar, S. (2018). Phase I/II dose-finding design for molecularly targeted agent: plateau determination using adaptive randomization. Statistical methods in medical research, 27(2), 466-479.

---

### Decision · Program_Chairs · 2021-09-27

**Decision:**

Accept (Poster)

**Comment:**

All reviewers have agreed that the paper is nicely written and presents a proper solution to the problem examined (given the fixes in the proof agreed with the authors in the discussion phase), and hence all of them recommend acceptance. On the other hand, there is also an agreement that the paper lacks proper motivation, as none of the examples mentioned by the authors (including the ones in the discussion) properly fit the framework provided. The paper would be an excellent fit to the conference with a proper motivation, and the authors are highly encouraged to demonstrate the existence of a realistic-looking problem where the framework is indeed applicable (otherwise this will remain a purely mathematical contribution).